# Study of the Morphology and Properties of Biocompatible Ca-P Coatings on Mg Alloy

**DOI:** 10.3390/ma13010002

**Published:** 2019-12-18

**Authors:** Katarzyna Cesarz-Andraczke, Ryszard Nowosielski, Marcin Basiaga, Rafał Babilas

**Affiliations:** 1Faculty of Mechanical Engineering, Silesian University of Technology, 44-100 Gliwice, Poland; Ryszard.Nowosielski@polsl.pl (R.N.); Rafal.Babilas@polsl.pl (R.B.); 2Faculty of Biomedical Engineering, Silesian University of Technology, 41-800 Zabrze, Poland; Marcin.Basiaga@polsl.pl

**Keywords:** Mg alloys, Ca-P coatings, hydrogen evolution, adhesion test

## Abstract

Magnesium alloys are considered as potential biomaterials for use in orthopedic implantology. The main barrier to the use of Mg alloys in medicine is their overly fast and irregular degradation in body fluids. The use of protective calcium phosphate coatings to increase the corrosion resistance of Mg alloy (AM50 alloy: 4 wt.% Al, 0.3 wt.% Mn, 0.2 wt.% Zn, rest Mg) was examined in this study. The scientific goal of the study was the assessment of the influence of calcium phosphate layer morphology on the corrosion process in Ringer’s solution. Modification of the coating morphology was obtained by changing the chemical composition of the phosphatizing bath using NaOH (NaAM50 sample) or ZnSO_4_ (ZnAM50 sample). In practice, a more dense and uniform coating could be obtained by the immersion of AM50 alloy in a solution containing ZnSO_4_ (ZnAM50 sample). In this study, an adhesion test performed on the ZnAM50 sample indicated that the critical load was 1.35 N. XRD phase analysis confirmed that the obtained coatings included dicalcium phosphate dihydrate (CaHPO_4_*2H_2_O). The coatings prepared on the NaAM50 and ZnAM50 samples are effective barriers against the progress of corrosion deeper into the substrate. After 120 h immersion in Ringer’s solution, the volume of the evolved hydrogen was 5.6 mL/cm^2^ for the NaAM50 and 3.4 mL/cm^2^ for the ZnAM50 sample.

## 1. Introduction

Magnesium alloys are the subject of many research papers in which they are considered as potential biomaterials. The main potential application of Mg alloys in medicine are orthopedic implants and materials for cardiovascular stents [1,2,3,4]. In the orthopedic implantology area, there has been some successful use of Mg alloys as compression screws. MAGNEZIX is the commercial name of biodegradable orthopedic screws for osteosynthetic applications [5].

However, the AM50 (4 wt.% Al, 0.3 wt.% Mn, 0.2 wt.% Zn, rest Mg) alloy is considered as a material for possible biomaterial use in implantology [6,7]. Unfortunately, uncoated AM50 alloy gave too high and unstable a degradation rate in physiological fluid [8]. The rapid degradation rate of the Mg alloy results in a loss of mechanical integrity in the physiological environment. In the case of Mg alloys, the high corrosion rate leads to a high hydrogen evolution rate. It should be noted that hydrogen gas is non-toxic and easily diffusible in the human body. However, excessive corrosion of Mg alloy can lead to the formation of gas bubbles in the surrounding soft tissue [9].

Witte et al. [10], who studied the degradation process of Mg alloys, concluded that the initially high degradation rate decreased with the formation of corrosion products. It should be noted that the corrosion products’ layer only temporarily improves protection against an increase in the rate of the corrosion process. Therefore, it is very important to investigate new materials or coatings for Mg alloys with stable corrosion rates and controlled corrosion properties in order to be suitable for clinical applications.

For controlling the degradation process of Mg alloys, many types of anticorrosion coatings have been widely reported in the literature. Until now, mainly ceramic coatings, such as oxides [11,12,13] and phosphates [14,15,16], have been considered for increasing the corrosion resistance of magnesium alloys. In addition, polymer coatings have been tested as resorbable coatings, including PLA, PLGA, and copolymers [17,18]. Composite coatings that increase the corrosion resistance of magnesium alloys, including oxide-phosphate [19,20] and ceramic-polymer [21,22] coatings, have been examined by several researchers.

The ideal protective coating includes components present in human bone, such as calcium phosphates (Ca-P). Ca-P coatings can be bioactive and accelerate bone formation processes. Anticorrosion coatings on Mg alloy surfaces should be mainly dense and uniform. The morphology of coatings is as equally important as their chemical composition.

In this study, Ca-P coatings on AM50 alloy were examined. The scientific goal of this work was an assessment of the influence of phosphate layer morphology on the corrosion process in Ringer’s solution. Modification of coating morphology was obtained by changing the chemical composition of the phosphatizing bath. Our chemical method for coating preparation is cost effective and easily converted to large-scale production [23]. NaOH and ZnSO_4_ were used for surface modification. In the present study, NaOH and ZnSO_4_ work as accelerators to form a dense and uniform coating when added to phosphatizing baths. The NaOH is used for alkali pretreatment to improve the corrosion resistance of Mg alloys [14], while ZnSO_4_ is used in phosphatizing baths for the deposition of Zn or ZnO on the Mg alloy’s surface to protect against any increase in the rate of corrosion.

## 2. Materials and Methods

The magnesium alloy AM50, used as the substrate for the Ca-P coatings, was produced by die casting. Chemical composition was determined by spectrometry method (by company, which produce Mg alloys). Chemical composition of AM50 alloy was 4 wt.% Al, 0.3 wt.% Mn, 0.2 wt.% Zn, rest Mg. The AM50 was cut into rectangular samples with dimensions of ~20 × 20 × 5 mm^3^.

A chemical method was used for the Ca-P coatings’ preparation, the process consisting of two stages. In the first step the substrate surface was preparation. The surface of the samples was polished by SiC paper with 220 and 600 grit size, cleaned using deionized water during 3 min and ethanol during 1 min, and then dried in air.For surface of the samplessubstrate the arithmetic average roughness profile, Ra measured by optic profilometer was 1.53 µm.

In the second step the phosphatizing bath was then prepared. The chemical composition of the phosphatizing bath was:0.05 M Ca(NO_3_)_2_·4H_2_O + 0.03 M NaH_2_PO_4_·2H_2_O + distilled water: sample named WAM500.05 M Ca(NO_3_)_2_·4H_2_O + 0.03 M NaH_2_PO_4_·2H_2_O + distilled water + 0.03 M ZnSO_4_·7H_2_O sample named ZnAM500.05 M Ca(NO_3_)_2_·4H_2_O + 0.03 M NaH_2_PO_4_·2H_2_O + distilled water + 0.03 M NaOH sample named NaAM50

Chemicals of analytical grade purity (produced by Chempur, Tarnowskie Góry, Poland) were used to prepare solutions for the all samples.

In the next step, the samples were immersed in a beaker containing the phosphatizing bath at room temperature over 24 h, and then the coated AM50 samples were dried in the air for 24 h at room temperature.

The determination of the morphology and chemical composition of Ca-P coatings on AM50 before and after immersion in Ringer’s solution was performed with a Supra 35 Carl Zeiss SEM with energy-dispersive X-ray spectroscopy (Jena, Germany) EDS/EDXA. The approximate thickness of the performed coatings was measured on cross section samples using scanning electron microscopy (Supra 35 Carl Zeiss, Jena, Germany). The X-ray diffraction measurements were performed with a Panalytical X’Pert Pro MPD diffractometer (Almelo, Netherlands) using filtered radiation from a cobalt-anode lamp (λKα = 0.179 nm), and a PIXcell 3D detector (Almelo, Netherlands) on the diffracted beam axis. The diffractograms were recorded over an angular range of 10–100° [2θ], with a step = 0.05°. The qualitative phase analyses of the substrate and coated samples before and after immersion in Ringer’s solution were performed using Panalytical High Score Plus software (Version 1.4) with the dedicated PAN-ICSD database. The immersion tests and measurements of the evolved hydrogen volume of the coated sample and substrate were carried out in Ringer’s solution (8.6 g/dm^3^ NaCl, 0.3 g/dm^3^ KCl, and 0.48 g/dm^3^ CaCl_2_*6H_2_O) over 120 h using the experimental station presented in the following references [24,25]. Immersion tests were performed on the experimental station performed by the authors, which is used to measure the evolved hydrogen. The experimental station consists of: a thermostat, a glass beaker with Ringer’s solution with a sample inside, a burette with a shut-off valve, a vessel with a bottom tube for balancing the pressure in the burette with atmospheric pressure, a venting valve, a silicone stopper. The thermostat maintained the temperature of 37 °C in the glass beaker throughout the immersion test. The burette was filled with Ringer’s solution and the shut-off valve was closed. The vent valve fitted in the silicone stopper after closing the valve ensured the absence of air bubbles in the sample vessel that could cause uncontrolled changes in the liquid level in the burette. The silicone stopper prevented the solution from evaporating in the dish. The evolved hydrogen gas is directed to the burette through the funnel, which in turn causes a decrease in the level of the solution in the burette. The changes of solution level in the burette after alignment with the level of solution from the vessel with the tube were performed at 24 h intervals. It should be noted that the experimental station was used to determine the volume of only evolved hydrogen. Some amount of the hydrogen gas is absorbed by magnesium alloys and this amount of the absorbed hydrogen has been undetermined. All studied samples were purified in acetone before immersion tests. The immersion tests was conducted during 120 h.

The corrosion behavior of uncoated and coated samples were studied by electrochemical tests using Autolab 302 N workstation controlled by NOVA software (Version 1.11). The corrosion studies were conducted in Ringer’s solution at 37 °C. The electrochemical tests were conducted using the three-electrode cell. The cell was equipped with a working electrode (sample), a reference electrode (AgCl electrode) and a counter electrode (platinum rod). The corrosion resistance of the studied samples was evaluated by recording of the open-circuit potential (*E*_OCP_) changes in a function of time and polarisation curves in the potential range *E*_OCP_−200 mV to *E*_OCP_+200 mV.

The adhesion tests of the Ca-P layer applied to the ZnAM50 sample were carried out by a scratch test using an open platform equipped with a CSM Micro-Combi-Tester (Peseux, Switzerland). The test was performed by making a crack using a penetrator (Rockwell diamond cone, Peseux, Switzerland), which was loaded with a gradual increase in the normal force. To assess a value of the critical force, L_c_, the friction force and friction coefficient were used (with microscopic observations, which are an integral part of the platform). The tests were carried out with an increasing loading force from 0.03 to 30 N, and the following operating parameters: loading speed 10 N/min, table travel speed 0.5 mm/min, and scratch length ~3mm.

In order to determine the surface wettability, the wetting angle θ were evaluated with the use of the Owens-Wendt method. The wettability angle measurement were performed using distilled water liquid(θ_w_) (by Poch S.A.) of the volume 1.5 μL, at room temperature (T = 23°C) at the test stand consisting of SURFTENS UNIVERSAL goniometer by OEG and a PC with Surftens software (Version 4.5). The values of surface free energy (SFE) assumed for the calculations, including their polar and dispersion components were respectively, for distilled water: γs = 51.0 mJ/m^2^ and γs = 21.8 mJ/m^2^.

## 3. Results

SEM images and EDS spectra of coatings performed on AM50 alloy surfaces are shown in Figure 1. The SEM image of the coating obtained by immersion in a bath included 0.05 M Ca(NO_3_)_2_·4H_2_O + 0.03 M NaH_2_PO_4_·2H_2_O + distilled water (WAM50 sample) showed petals-like morphology (Figure 1a). The petals have smooth edges and are arranged in several directions. The coating obtained by immersion in a phosphatizing bath with ZnSO_4_ addition (ZnAM50 sample) consisted of petals (Figure 1c). The coating was compact with areas contained agglomerates of petals. The SEM image of the coating obtained by immersion in a bath with NaOH addition (NaAM50 sample) showed plate-like morphology (Figure 1e). The plates were arranged in several directions. The EDS analysis of WAM50 and ZnAM50 samples indicated the presence of Ca, P and O (Figure 1b,d) mainly. The EDS analysis of NaAM50 sample indicated the presence of Ca, P, O and Mg (Figure 1f).

In Figure 2, the cross sections of the WAM50, ZnAM50 and NaAM50 samples are presented. For selected areas the approximate thickness of the obtained coatings was about ≈14, 11,12 µm respectively. The coatings thickness is only estimated for selected areas. It should be noted that the thickness of the all studied coatings varied in size depending on the selected area.

X-ray diffraction patterns of coated samples and substrate AM50 are shown in Figure 3. The phase composition analysis of the AM50 substrate indicated the presence of an α-Mg phase. In the case of the WAM50, NaAM50 and ZnAM50 samples, the XRD patterns peaks refers to dicalcium phosphate dihydrate (CaHPO_4_*2H_2_O).

Figure 4 shows the results of immersion tests in Ringer’s solution at 37 °C. After 120 h immersion, the volume of evolved hydrogen (Figure 4a) was about 8.7 mL/cm^2^, 4.0 mL/cm^2^, 5.6 mL/cm^2^, and 3.4 mL/cm^2^ for AM50, WAM50, NaAM50, and ZnAM50, respectively. The hydrogen evolution rate (Figure 4b) for the ZnAM50 sample was stable at ≈0.2 mL/cm^2^/h. For the other two samples, the hydrogen evolution rate decreased with immersion time.

The electrochemical tests results measured in Ringer’s solution at 37 °C for non-coated AM50 sample and coated WAM50, NaAM50, ZnAAM50 samples are presented in Figure 5. The more positive value of the E_OCP_ was detected for WAM50, NaAM50, ZnAAM50 samples rather than AM50 sample (Figure 5a). After 1 h measurement for AM50, NaAM50, WAM50, ZnAAM50 samples E_OCP_ is 1.53, 1.50, 1.49, 1.44V respectively. Similarly, slightly changes of the E_OCP_ was noticed for uncoated AM50 sample. The highest fluctuations of the E_OCP_ were noticed for the ZnAM50 sample.

The WAM50, NaAM50, ZnAAM50 samples have more positive value of E_corr_ than non-coated AM50 sample (Figure 5b). For NaAM50, WAM50, ZnAAM50 samples E_corr_ is 1.50, 1.48, 1.46 V respectively. For non-coated AM50 sample E_corr_ is 1.50 V. The cathodic part of potentiodynamic curve determined for the WAM50, NaAM50, ZnAAM50 samples is located in a low current range, which indicated a low cathodic activity.

In order to determine the surface wettability, the wetting angle θ were evaluated for substrate (uncoated AM50 alloy) and for ZnAM50 sample. Hydrophilic surface, but showing low wettability, was observed for ZnAM50 sample (θav = 83.95°). The application of the Ca-P coating reduced the contact angle (θav = 66.15°)but all the time that was hydrophilic surface.

The results of the adhesion test of the CaHPO_4_*2H_2_O coating on the AM50 alloy sample carried out by the scratch method are presented in Figure 6. Based on the scratch test result, it was found that the critical load that was a measure of adhesion for the ZnAM50 sample was 1.35 N.

The surface morphology and EDX chemical composition analysis results for the studied samples are shown in Figure 7. The surfaces of all samples were covered by corrosion product layers with visible small microcracks (Figure 7a,c,e). However, EDX analysis (Figure 7b,d,f) revealed a significant difference in the chemical composition of the studied surfaces corrosion products. From the surface of WAM50 and ZnAM50 samples, high intensity peaks from O, P, and Ca were identified. Besides, EDX analysis indicated the presence of Na, Cl, and Mg. In turn, from the surface of NaAM50 sample, high intensity peaks from O and Mg were identified. In addition, EDX analysis indicated the presence of Na, Cl, P, Ca, and Al.

X-ray diffraction patterns of the coated samples and substrate after 120 h immersion in Ringer’s solution are shown in Figure 8. The phase composition analysis of the AM50 substrate indicated the presence of Mg(OH)_2_ and an α-Mg phase. After immersion tests, the surfaces of the WAM50, ZnAM50 and NaAM50 samples were covered by Mg(OH)_2_ and CaHPO_4_*2H_2_O.

## 4. Discussion

In this study Ca-P coatings on AM50 alloy were performed. Study of the morphology of the obtained coatings indicated petals (Figure 1a,c) or plate-like structures (Figure 1e). Addition of NaOH to the phosphatizing bath caused a modification of the coating morphology. The structure of this layer is more developed than in the case of the layer formed in a phosphatizing bath with ZnSO_4_. The shape and orientation of elements, which included Ca-P coating (Figure 1b) obtained in bath with ZnSO_4_ addition caused that this coating are more dense than Ca-P coating obtained in bath with NaOH addition included plates elements (Figure 1e). Chemical composition analysis (EDS, Figure 1f) of the NaAM50 sample identified a small peak from Mg. It could be possible that orientation elements of the Ca-P layer on the NaAM50 sample were unevenly distributed and formed areas that revealed the substrate. Anawati et al. [26] proved that, alkali treatment induced the formation of a nano-size platelet Mg(OH)_2_ layer on the film surfaces that drastically enlarged the effective surface area for the precipitation of apatite. In this work, instead of doing alkali treatment, NaOH added to phosphatizing bath. In the system brushite-NaOH is possible transformation brushite to hydroxyapatite. Furutaka et al. [27] described characteristic reaction processes in the system brushite-NaOH solution. In this work results of studies indicate that NaOH addition to phosphatizing bath caused morphological changes of DCPD (brushite). From petals morphology of DCPD (Figure 1a,c) was obtained plate DCPD morphology (Figure 1f) after added NaOH to phosphatizing bath.

The ZnSO_4_ was used like Zn ion source and added to phosphatazing bath. Unfortunately, in this work results of surface studies did not indicate a formation of Zn in Ca-P layer on ZnAM50 sample. In comparison to sample without ZnSO_4_ addition in phospathizing bath the ZnSO_4_ addition caused growth of Ca-P coating (Figure 1c). Similar elements of both Ca-P coatings proved this growth. Marked on inset of Figure 1a,c.

The XRD phase analysis (Figure 3) confirmed that the obtained coatings included dicalcium phosphatizing dihydrate (CaHPO_4_*2H_2_O). DCPD is characterized by its biocompatibility and osteoconductive properties [28]. Dicalcium phosphate dihydrate (DCPD) is widely used as a coating for biomedical materials [29,30]. 

Besides, both coatings caused hydrogen evolution decrease in comparison to the AM50 substrate. After 120 h in Ringer’s solution, the volume of evolved hydrogen (Figure 4a) was 5.6 mL/cm^2^ for NaAM50 and 3.4 mL/cm^2^ for ZnAM50. The coatings prepared on the NaAM50 and ZnAM50 sample are effective barriers against the progress of corrosion deeper into the substrate. In paper [31], an AZ91 alloy with a DCPD coating during 120 h immersion in simulated body fluid evolved about 5 mL/cm^2^/24 h. For comparison, a substrate of AZ91 during 120 h immersion in simulated body fluid evolved about 40 mL/cm^2^/24 h. Kajanek et al. [32] obtained better corrosion properties from a magnesium alloy with a DCPD coating than an uncoated substrate, the corrosion rate decreasing four times in comparison to the uncoated substrate. In addition, in the present study, the hydrogen evolution rate (Figure 4b) for the ZnAM50 sample was steady at≈0.2 mL/cm^2^/h. For the other two samples, the hydrogen evolution rate decreased with immersion time. This steady hydrogen evolution rate of the ZnAM50 sample may be a result of the coating morphology on the sample’s surface, as the coating on this sample is more dense and uniform.

Taking into account more positive value E_corr_, E_OCP_ and location in lower current range of potentiodynamic curves for coated samples in comparison of non coated sample (Figure 5a,b) it can be found, that Ca-P coatings obtained in this work improve corrosion resistance in Ringer’s solution. It should be noted, that the lower cathodic activity (related to lower hydrogen evolution) correlates with the results of immersion tests for studied samples. The DCPD coatings on WAM50, NaAM50, ZnAAM50 samples caused that values of the E_corr_ are moved to more positive potential values to the value of non-coated AM50 sample. In work [32] the DCPD coated sample have reached more positive E_corr_(−1403 mV) to the value of non-coated sample (−1554 mV). Similarly, in work [33], the electrochemical characteristics of AZ31 alloy in 0.9% NaCl solution were improved by depositing a DCPD coating. The polarization resistance (R_p_) decreased by nearly ten times in comparison to an uncoated substrate sample.

In this study, an adhesion test was performed for the ZnAM50 sample in order to achieve better protection properties (Figure 6). It was found that the critical load that is a measure of adhesion for the ZnAM50 sample was 1.35 N. In paper [34], the recorded critical loads (for initial delamination) for Ca-P coatings formed at 19 °C and 28 °C were 0.2 N and 0.9 N, respectively. It should be noted that in paper [34] the critical load values were obtained for Ca-P coatings on a poly(carbonate urethane) substrate.

The results of phase analysis (Figure 8) of the corrosion products formed after immersion tests on ZnAM50 and NaAM50 showed that the samples’ surfaces were covered by Mg(OH)_2_ and CaHPO_4_*2H_2_O. For the NaAM50 sample, more peaks from Mg(OH)_2_ were identified than for the ZnAM50 sample. These results correlate with the EDS chemical composition results for the corrosion products of both the NaAM50 and ZnAM50 samples (Figure 7c,e). Figure 7c shows peaks of Ca, P, and O with high intensity, which indicate the formation of a DCPD coating on ZnAM50 sample. In turn, Figure 7f shows peaks of Mg and O with high intensity, which indicate the existence of MgO on NaAM50 sample. There are also Cl and Na peaks, which are residues of Ringer’s solution.

It can be assumed that the DCPD coating has better protection properties against the progress of the corrosion process than Mg(OH)_2_. The greatest influence on the protection properties of coatings is probably their solubility in aqueous fluids. Due its chemical and structural similarities with human bone, DCPD has low solubility in human body fluids or blood plasma [28].

Living organisms react to biomaterials through their surface layer. The analysis of biomaterial surface coating regarding their wettability allows evaluating the possible reaction of organism cells. The ZnAM50 sample surface is hydrophilic. Higher wettability, and thus lower wetting angle, favour cell adhesion to biomaterial surface, which is important in the case of orthopaedic implants as itinfluences the process of tissue—implant integration. On the other hand, lower wettability—and thus a higher value of wetting angle, is vital in the case of such clinical applications as, for example, elements of cardiacvalvesor devices used in dialys is where the lowest protein absorption is desired as it cuts down blood coagulation.

Morphological modification by wet chemical methods of the calcium phosphate (Ca-P) coatings obtained on the surface of magnesium alloys is an unexplored subject in the literature. Many research teams focus on the preparation method of Ca-P coatings and investigating their properties [35]. In this study, the subject of our research is the possibility of modifying the calcium phosphate coating by using chemical reagents to achieve morphology that provides protection against the corrosion of the Mg alloy for the period of bone union. In the literature we can find a few studies about the modification of coatings on Mg-based alloys. For example, the authors of paper [33] tried to modify calcium phosphate coatings by using C_6_H_4_O_5_NSNa and Na_2_MoO_4_. In this work, for the first time NaOH and ZnSO_4_ have been used to modify Ca-P coatings’ morphology by a wet chemical method on the surface of an AM50 alloy. In work [36], the modification of the calcium phosphate coatings was based on a change in electrodeposition parameters. It should be noted that wet chemical deposition is a less expensive method than electrodeposition. A chemical deposition method was used in [37] to obtain a calcium phosphate coating on an AZ31 alloy. In paper [37], varied processing temperatures were used to obtain a different microstructure, phase, and morphology of Ca-P coatings. With the aim of modifying Ca-P coatings’ morphology and properties, pulse electrodeposition methods were used [38]. Consequently, a Si-doped calcium phosphate coating was obtained.

## 5. Conclusions

In this work the Ca-P coatings on magnesium alloys by simple wet chemical method were successfully prepared. Results of corrosion studies indicated that obtained Ca-P coatings are effective barriers against the progress of corrosion deeper into the substrate. In addition, the chemical composition analysis of prepared coatings included dicalcium phosphate dihydrate (DCPD). As biomaterial, the dicalcium phosphate dihydrate characterize biocompatibility and osteoconductive properties. It could be possible, that coatings that include dicalcium phosphate dihydrate may have bioactive functions. These coatings probably may be potential good protection against irregular and too fast degradation process of magnesium alloys considered as resorbable biomaterial for implantology.

## Figures and Tables

**Figure 1 materials-13-00002-f001:**
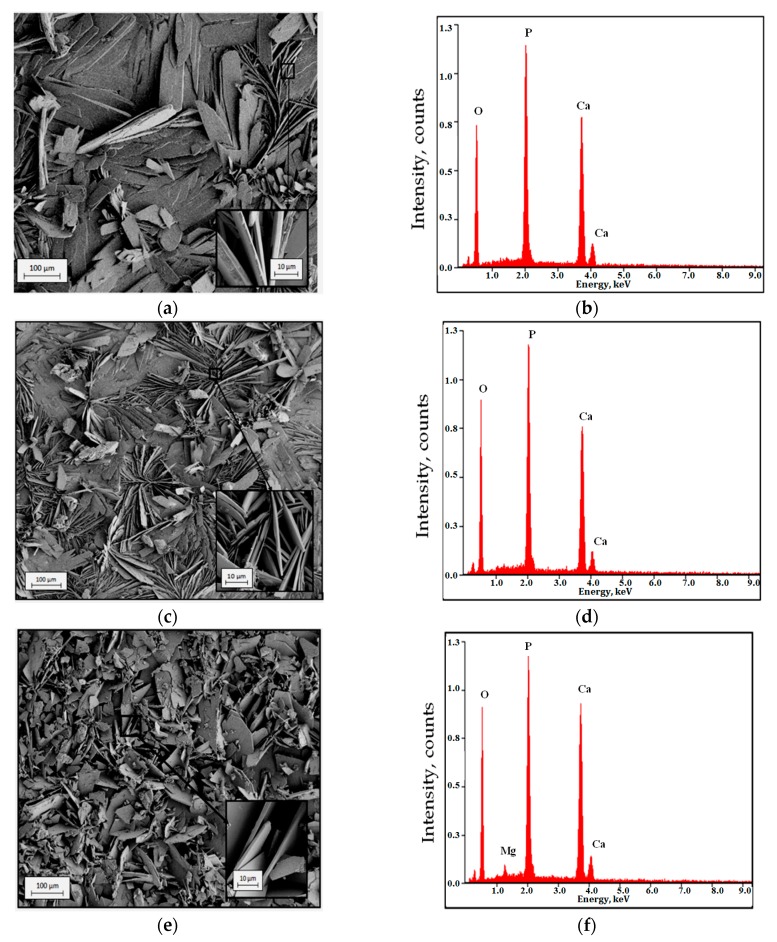
SEM images of surface morphology and EDX spectra of: (**a**,**b**) WAM50, (**c**,**d**) ZnAM50, (**e**,**f**) NaAM50 samples.

**Figure 2 materials-13-00002-f002:**
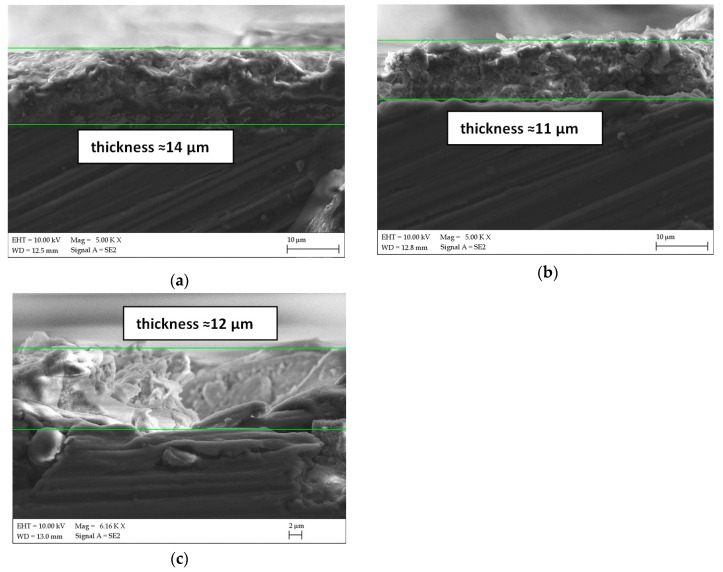
Cross section SEM images of: (**a**) WAM50, (**b**) ZnAM50, (**c**) NaAM50 samples with approximate layer thickness.

**Figure 3 materials-13-00002-f003:**
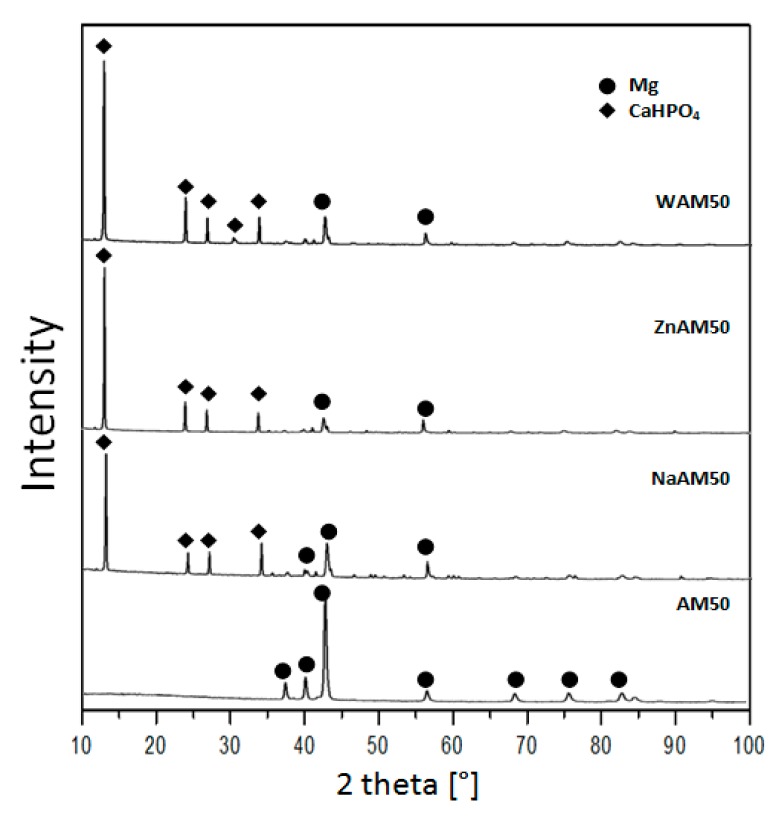
XRD patterns of substrate and coated samples.

**Figure 4 materials-13-00002-f004:**
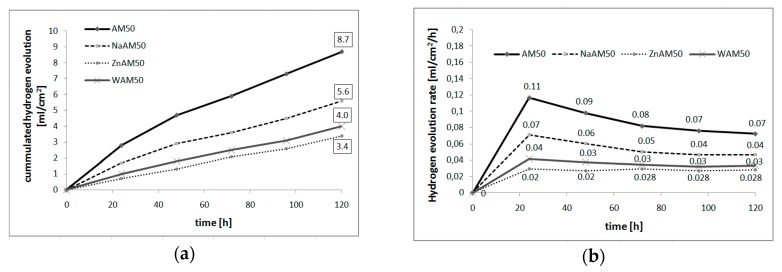
Results of immersion tests of studied samples in Ringer’s solution at 37°C: (**a**) cumulated hydrogen evolution, (**b**) hydrogen evolution rate.

**Figure 5 materials-13-00002-f005:**
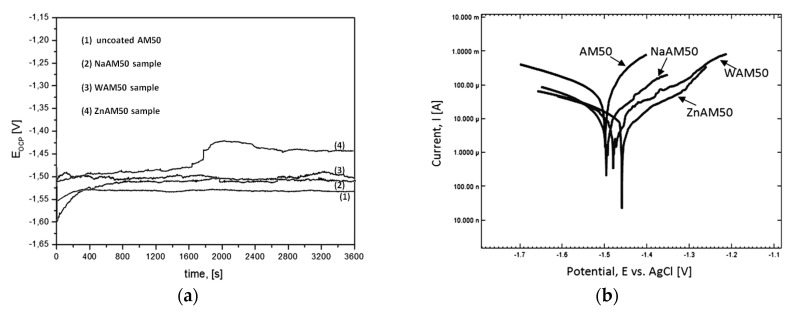
Results of electrochemical tests of studied samples in Ringer’s solution at 37 °C: (**a**) changes of open-circuit potential in function of time, (**b**) potentiodynamic curves.

**Figure 6 materials-13-00002-f006:**
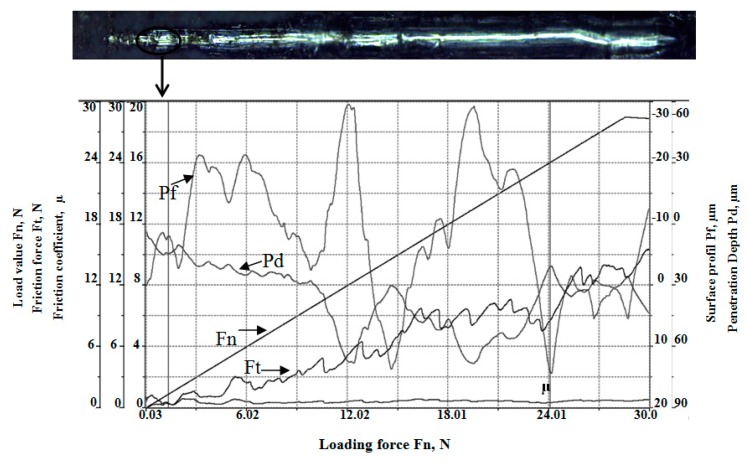
Scratchand results of adhesion test of the dicalcium phosphate dihydrate coating on AM50 alloy sample carried out by the scratch method.

**Figure 7 materials-13-00002-f007:**
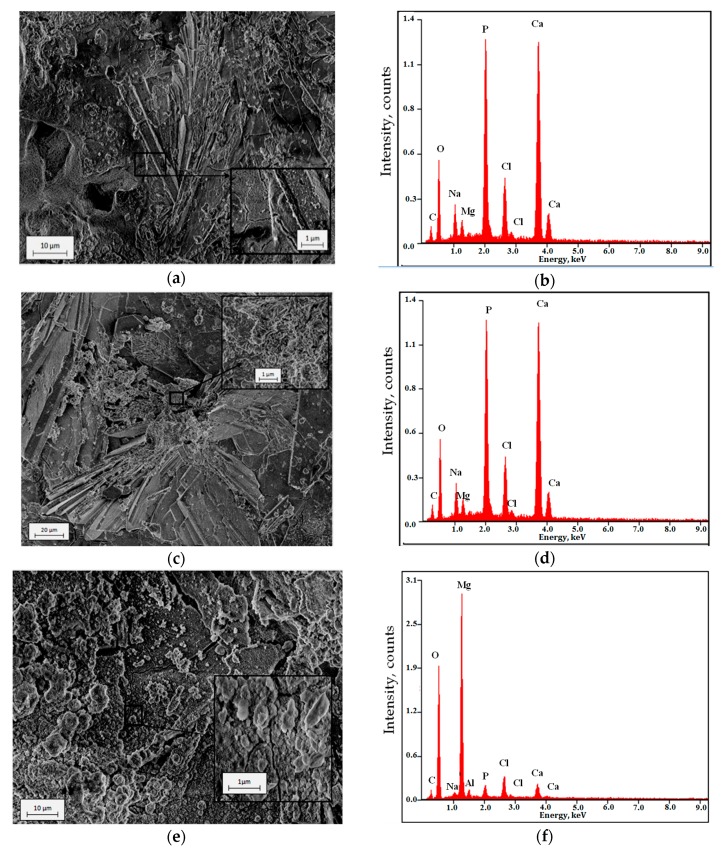
Surface morphology after 120 immersion in Ringer’solution at 37 °C and EDX spectra of: *(***a**,**b**) WAM50, *(***c**,**d**) ZnAM50, *(***e**,**f**) NaAM50 samples.

**Figure 8 materials-13-00002-f008:**
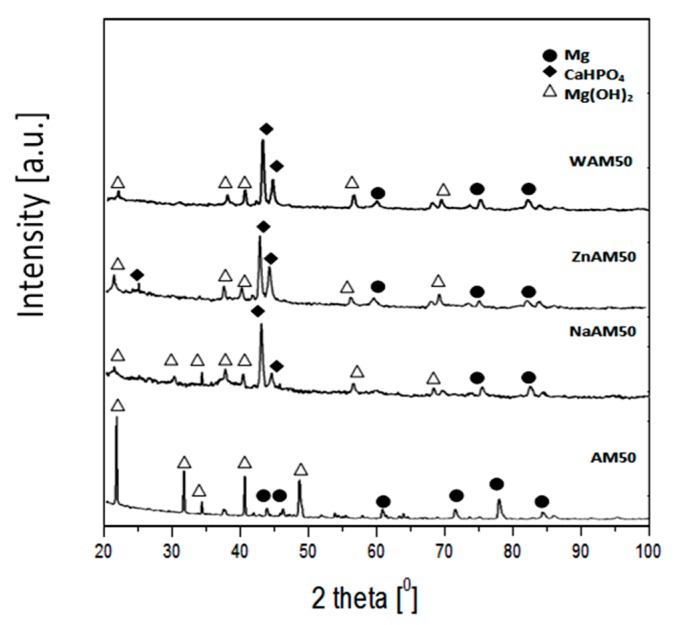
XRD patterns of substrate and coated samples after 120 h immersion in Ringer’s solution at 37 °C.

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
