# Peer review of "Study of the Morphology and Properties of Biocompatible Ca-P Coatings on Mg Alloy"

_materials, 2019, doi:10.3390/ma13010002_

Round 1

Reviewer 1 Report

The topic of studying the morphology and properties of biocompatible Ca-P coatings on Mg alloy is very interesting. However, the authors do not provide significant insights on the explanation of how NaOH and ZnSO4 influence the morphology and why they have different influence.

2. The authors should also compare the Ca-P coating made from Phosphatizing bath without NaOH and ZnSO4 to the ones with NaOH or ZnSO4.

3. The authors should explain why Coating made with ZnSO4 are more dense than coatings made with NaOH.

4. Besides hydrogen measurement, more corrosion tests such as electrochemical tests should be conducted.

Author Response

First of all, authors would like to say thanks to Reviewer for remarks and suggestions. Authors are very grateful for helpful comments in case of the increase of the presented paper readability and quality. The changes were marked by using red colour in the revised manuscript.

1. The topic of studying the morphology and properties of biocompatible Ca-P coatings on Mg alloy is very interesting. However, the authors do not provide significant insights on the explanation of how NaOH and ZnSO4 influence the morphology and why they have different influence.

A: Anawati et al [x]  proved that,  alkali treatment induced the formation of a nano-size platelet Mg(OH)2 layer on the film surfaces that drastically enlarged the effective surface area for the precipitation of apatite. In this work, instead of doing alkali treatment, NaOH added to phosphatizing bath. In the system brushite- NaOH is possible transformation brushite to hydroxyapatite. Furutaka et al [y] described characteristic reaction processes in the system brushite–NaOH solution.

In this work results of studies indicate that NaOH addition to phosphatizing bath caused morphological changes of DCPD (brushite). From petals morphology of DCPD (Fig.1a, Fig.1b) was obtained plate morphology of DCPD (Fig.1e) after added NaOH to phosphatizing bath.

The ZnSO4 was used like Zn ion source and added to phosphatazing bath. Unfortunately, in this work results of surface studies did not indicate a formation of Zn in Ca-P layer on ZnAM50 sample. In comparison to sample without ZnSO4 addition in phospathizing bath (Fig.1a) the ZnSO4 addition caused growth of Ca-P coating (Fig.1b). Similar elements of both Ca-P coatings proved this growth. Marked on inset of Fig. 1a and Fig. 1b

 [x] H. Asoh, S. Ono Anawati, Enhanced uniformity of apatite coating on a PEO film

formed on AZ31 Mg alloy by an alkali pretreatment, Surf. Coating. Technol.

272 (2015) 182e189.

 [y]K. Furutaka, H. Monma, T. Okura, S. Takahashi,Characteristic reaction processes in the system brushite–NaOH solution, Journal of the European Ceramic Society

Volume 26, Issues 4–5, 2006, Pages 543-547

 In this work studies results showed that NaOH addition to phosphatizing bath caused morphological changes of DCPD (brushite). From petals morphology of DCPD (Fig.1a, Fig.1b) we obtained plate morphology of DCPD (Fig.1e) after added NaOH to phosphatizing bath. In comparison to sample without ZnSO4 addition in phospathizing bath (Fig.1a) the ZnSO4 addition caused growth of Ca-P coating (Fig.1b). Similar elements of both Ca-P coatings proved this growth. This elements marked on zoom Fig1a and Fig.1b.

The authors should also compare the Ca-P coating made from Phosphatizing bath without NaOH and ZnSO4 to the ones with NaOH or ZnSO4.

A: The results of sample without NaOH and ZnSO4 in phosphatizing bath were added to work and compare to the Ca-P coating with NaOH and ZnSO4 additions.

Results of WAM50 ( sample with coating obtained in phosphatizing bath without ZnSO4 and NaOH additions) studies are in Fig.1a,c ; Fig.2a ; Fig.3 ; Fig.4 ; Fig.5, Fig.7a,c ; Fig.8

The authors should explain why Coating made with ZnSO4 are more dense than coatings made with NaOH

A:The shape and orientation of elements(petals), which included Ca-P coating (Fig.1b) obtained in bath with ZnSO4 addition caused that this coating are more dense than Ca-P coating obtained in bath with NaOH addition included plates elements (Fig.1e)

Besides hydrogen measurement, more corrosion tests such as electrochemical tests should be conducted.

A: The electrochemical tests was made for coated samples and substrate. Results of electrochemical tests were added. Taking into account more positive value Ecorr, EOCP and location in lower current range of potentiodynamic curves for coated samples in comparison of non coated sample (Fig.5a,b) it can be found, that Ca-P coatings obtained in this work improve corrosion resistance in Ringer's solution. It should be noted, that the lower cathodic activity (related to lower hydrogen evolution) correlates with the results of immersion tests for studied samples. The DCPD coatings on WAM50, NaAM50, ZnAAM50 samples caused that values of the Ecorr are moved to more positive potential values to the value of non-coated AM50 sample. In work [31] the DCPD coated sample have reached more positive Ecorr (-1403 mV) to the value of non-coated sample (-1554 mV). Similarly, in paper [32], the electrochemical characteristics of AZ31 alloy in 0.9% NaCl solution were improved by depositing a DCPD coating. The polarization resistance (Rp) decreased by nearly ten times in comparison to an uncoated substrate sample.

Reviewer 2 Report

Dear authors,

This is a review on the paper entitled “Study of the morphology and properties of 2 biocompatible Ca-P coatings on Mg alloy” by Katarzyna Cesarz-Andraczke, Ryszard Nowosielski, Marcin Basiaga and Rafał Babilas, submitted for Materials Journal, manuscript number materials-657524.

The authors want to analyse the use of protective calcium phosphate coatings to increase the corrosion resistance of Mg alloys.

The paper fits to the journal field wishing presenting a study on the assessment of the influence of phosphate layer morphology on the corrosion process in Ringer's solution.

However, the article has some shortcomings.

Abstract

Needs to be reformulated so that one could understand all the information. Eg. The authors present in the abstract AM50 alloy, ZnAM50, NaAM50 but without any explanation. Also “the obtained coatings included dicalcium phosphate dihydrate (CaHPO4*2H2O), and therefore the coatings showed bioactive functions” - presenting this way in the abstract is not quite sufficient to have bioactive functions. The phrase “Dicalcium phosphate dihydrate is characterized by its biocompatibility and osteoconductive properties” – does not belong to the abstract. The last phrase shoul also be reformulated.

The introduction part

is quite good, minor English control is required.

The experimental procedure

Materials

the exact chemical composition and method of determination of the AM50 alloy should be provided.

Sample Preparation

needs more description – especially regarding the surface to be coated – roughness, surface sample preparation, morphology eso. Line 79 and 80– ZnAM50 and NaAM50 denomination of the samples are inserted but with no explanation – the explanation is presented starting with line 108, so you should insert it here.

Methods

each method and the set-up used in the research must be completely described. Also can you describe more the adhesion test (can you also provide a reference?)

Results

Figure 1 – SEM images – please use the insets as regular forms (circle, square and not ellipse – it looks like distorted images); the EDS spectre (fig. 1 c and d) should have axis defined properly. Also line 112 you present only Ca, P and O elements but EDS spectra from fig 1, d shows also Mg. Figure 2 caption must be reformulated. Also – the results are badly presented. The presented measurements cannot remain in this form. There exist proper methods for layer thickness measurement and you must use one of these or you can measure by your method, but in a proper manner, and please use the decimals only if you confide the numbers. (eg. https://dl.asminternational.org/handbooks/book/20/chapter-abstract/289380/Film-Thickness-Measurements-Using-Optical?redirectedFromFulltext=true ) Line 122-124 phrase must be reformulated since is not right. Figure 5 – is very hard to read even in colour mode – colours are badly chosen. Also inserting some arrows would be good. Figure 6 – same comments as Figure 1

Discussions

needs more attention – and must discuss all the findings compared to other findings from different research. What did you mean with the next formulation - “The coating on the NaAM50 sample is characterized by more diversified thickness than that 225 on the ZnAM50 sample”?.

Conclusion

needs to be reformulated and present no numbering. Lines 225-226 belong to the results chapter Lines 228-229 belong to the discussion chapter. Here you present a completely different thing and correct compared to the phrase from abstract by using the word POSSIBLE. Partly lines 232 – 236 belong to the results chapter.

References

No comments

Author Response

First of all, authors would like to say thanks to Reviewer for remarks and suggestions. Authors are very grateful for helpful comments in case of the increase of the presented paper readability and quality. The changes were marked by using red colour in the revised manuscript.

However, the article has some shortcomings.

Abstract

Needs to be reformulated so that one could understand all the information. Eg. The authors present in the abstract AM50 alloy, ZnAM50, NaAM50 but without any explanation. Also “the obtained coatings included dicalcium phosphate dihydrate (CaHPO4*2H2O), and therefore the coatings showed bioactive functions” - presenting this way in the abstract is not quite sufficient to have bioactive functions. The phrase “Dicalcium phosphate dihydrate is characterized by its biocompatibility and osteoconductive properties” – does not belong to the abstract. The last phrase shoul also be reformulated.

 A: Abstract was reformulated.The changes were marked by using red colour in the revised manuscript.

The introduction partis quite good, minor English control is required.

A: The work has been proofread by a native speaker of English

The experimental procedure

Materialsthe exact chemical composition and method of determination of the AM50 alloy should be provided.

A: Information about AM50 alloy were added. Chemical composition was determined by spectrometry method by company, which produce Mg alloys. Chemical composition of AM50 alloy was 4 %wt Al, 0.3 %wt Mn, 0.2 %wt Zn, rest Mg. The changes were marked by using red colour in the revised manuscript.

Sample Preparation

needs more description – especially regarding the surface to be coated – roughness, surface sample preparation, morphology eso. Line 79 and 80– ZnAM50 and NaAM50 denomination of the samples are inserted but with no explanation – the explanation is presented starting with line 108, so you should insert it here.

 A: Description has been completed.  The surface of the samples was polished by SiC paper with a grit size of 220 and 600, cleaned using deionized water during 3 minutes and ethanol during 1 minute , and then dried in air. For surface of the samples substrate the arithmetic average roughness profile, Ra measured by optic profilometer was 1,53 µm

Methods

each method and the set-up used in the research must be completely described. Also can you describe more the adhesion test (can you also provide a reference?)

 A: Description has been completed.  I don't make a studies on a reference sample. The test was performed by making a crack using a penetrator (Rockwell diamond cone), which was loaded with a gradual increase in the normal force. To assess a value of the critical force, Lc, the friction force and friction coefficient were used, along with microscopic observations, which are an integral part of the platform. The tests were carried out with an increasing loading force from 0.03 to 30N, and the following operating parameters: loading speed 10N/min, table travel speed 0.5mm/min, and scratch length ~ 3mm.

Results

Figure 1 – SEM images – please use the insets as regular forms (circle, square and not ellipse – it looks like distorted images); the EDS spectre (fig. 1 c and d) should have axis defined properly. Also line 112 you present only Ca, P and O elements but EDS spectra from fig 1, d shows also Mg. Figure 2 caption must be reformulated. Also – the results are badly presented. The presented measurements cannot remain in this form. There exist proper methods for layer thickness measurement and you must use one of these or you can measure by your method, but in a proper manner, and please use the decimals only if you confide the numbers. (eg. https://dl.asminternational.org/handbooks/book/20/chapter-abstract/289380/Film-Thickness-Measurements-Using-Optical?redirectedFromFulltext=true ) Line 122-124 phrase must be reformulated since is not right. Figure 5 – is very hard to read even in colour mode – colours are badly chosen. Also inserting some arrows would be good. Figure 6 – same comments as Figure 1

A: Figures have been corrected according to the reviewer's recommendations.

The purpose of thickness testing was approximate estimation of thickness. The authors wanted to estimate the thickness scale, not a specific value. Several areas for measurement were selected. One is presented in the paper for each tested layer. Light microscope results have been removed. Thickness measurement were carried out on the SEM microscope.

 Discussions

needs more attention – and must discuss all the findings compared to other findings from different research. What did you mean with the next formulation - “The coating on the NaAM50 sample is characterized by more diversified thickness than that 225 on the ZnAM50 sample”?.

 A: Discussion sectionhave been corrected according to the reviewer's recommendations. The changes were marked by using red colour in the revised manuscript. I would like to say that more diversified thickness  means  that the thickness of the all studied coatings varied in size depending on the selected area. For selected areas the approximate thickness of the obtained coatings was about 14, 11,12 µm respectively. The coatings thickness is only estimated for selected areas.

Conclusion

needs to be reformulated and present no numbering. Lines 225-226 belong to the results chapter Lines 228-229 belong to the discussion chapter. Here you present a completely different thing and correct compared to the phrase from abstract by using the word POSSIBLE. Partly lines 232 – 236 belong to the results chapter.

A: Conclusions section have been corrected according to the reviewer's recommendations.

Reviewer 3 Report

The manuscript is written in very good level, but the theme is not focused in this journal and for this manuscript will be better submission in the another journal, I recommend publishing this paper in other journal, for example Coatings, Coatings

Instrumental method - operating parameters, precision, accuracy and uncertainties of methods, QA/QC procedure are missed.

Can you have quantified the amount of corrosion products, and can you added purity nad source of used chemicals.

Please control formatting of text and unify in whole document.

Author Response

First of all, authors would like to say thanks to Reviewer for remarks and suggestions. Authors are very grateful for helpful comments in case of the increase of the presented paper readability and quality. The changes were marked by using red colour in the revised manuscript.

The manuscript is written in very good level, but the theme is not focused in this journal and for this manuscript will be better submission in the another journal, I recommend publishing this paper in other journal, for example Coatings, Coatings

Instrumental method - operating parameters, precision, accuracy and uncertainties of methods, QA/QC procedure are missed.

A: Immersion tests were performed on the experimental station performed by the authors, which is used to measure the released hydrogen. The experimental station consists of: a thermostat, a glass beaker with Ringer's solution with a sample inside, a burette with a shut-off valve, a vessel with a bottom tube for balancing the pressure in the burette with atmospheric pressure, a venting valve, a silicone plug. The thermostat maintained the temperature of 37 C in the glass beaker throughout the immersion test. The burette was filled with Ringer's solution and the shut-off valve closed. The vent valve fitted in the silicone plug after closing the valve ensured the absence of air bubbles in the sample vessel that could cause uncontrolled changes in the liquid level in the burette. The silicone stopper prevented the solution from evaporating in the dish. The released hydrogen gas is directed to the burette through the funnel, which in turn causes a decrease in the level of the solution in the burette. Readings of changes in the level of liquid in the burette after alignment with the level of liquid from the vessel with the tube were made at 24 hour intervals. It should be noted that the experimental station is used to determine the volume of only released hydrogen. Part of the hydrogen gas is absorbed by magnesium alloys and this part of the absorbed hydrogen has not been determined. All samples were purified in acetone before immersion testing. The immersion tests was provide by 120hours.

2.Can you have quantified the amount of corrosion products, and can you added purity nad source of used chemicals.

A: To determine the amount of corrosion products it is necessary to perform tests, e.g. ICP EOS (solution test) or sample weight loss test. We have no opportunity to conduct ICP measurements. Performing them now will be subject to an error, because such tests should be performed immediately after removing the sample from the solution. All chemicals were of analytical grade purity ( produced by Chempur) were used to prepare solutions for the all samples.

3.Please control formatting of text and unify in whole document.

A:Format of text  have been corrected according to the reviewer's recommendations

Reviewer 4 Report

The paper written by CA. Katarzyna et al. illustrate a simple method for the coating of magnesium alloys in order to overcome the main barrier presented by this material for orthopedic implantology. The obtained material was tested in terms of morphology and adhesion properties.

The paper is well written and the results are has a good interpretation. Thus, I recommend the publication of this work in Materials Journal after a few remarks:

Since the main problem of magnesium alloys is the fast and irregular degradation into the fluids after implantation that was also mentioned by the authors in the current paper, I suggest the authors to perform also degradation studies on the as-synthesized material. Also, the hydrophilic character of the materials which will be further used as the implantable scaffold is a very important aspect that the authors should take into consideration. Considering this, I suggest the authors to perform contact angle measurements.  

Author Response

First of all, authors would like to say thanks to Reviewer for remarks and suggestions. Authors are very grateful for helpful comments in case of the increase of the presented paper readability and quality. The changes were marked by using red colour in the revised manuscript.

The paper is well written and the results are has a good interpretation. Thus, I recommend the publication of this work in Materials Journal after a few remarks:

Since the main problem of magnesium alloys is the fast and irregular degradation into the fluids after implantation that was also mentioned by the authors in the current paper, I suggest the authors to perform also degradation studies on the as-synthesized material. Also, the hydrophilic character of the materials which will be further used as the implantable scaffold is a very important aspect that the authors should take into consideration. Considering this, I suggest the authors to perform contact angle measurements. 

A: Authors made electrochemical tests of substrate and coated samples. Results of linear polarization and determined open circuit potential allows to  determined corrosion behavior and simulate/predict course of degradation process. Taking into account more positive value Ecorr, EOCP and location in lower current range of potentiodynamic curves for coated samples in comparison of non coated sample (Fig.5a,b) it can be found, that Ca-P coatings obtained in this work improve corrosion resistance in Ringer's solution. It should be noted, that the lower cathodic activity (related to lower hydrogen evolution) correlates with the results of immersion tests for studied samples. The DCPD coatings on WAM50, NaAM50, ZnAAM50 samples caused that values of the Ecorr are moved to more positive potential values to the value of non-coated AM50 sample. In work [31] the DCPD coated sample have reached more positive Ecorr (-1403 mV) to the value of non-coated sample (-1554 mV). Similarly, in paper [32], the electrochemical characteristics of AZ31 alloy in 0.9% NaCl solution were improved by depositing a DCPD coating. The polarization resistance (Rp) decreased by nearly ten times in comparison to an uncoated substrate sample.

 Authors perform also contact angle measurements. In order to determine the surface wettability, the wetting angle θ were evaluated for substrate (uncoated AM50 alloy) and for ZnAM50 sample. Hydrophilic surface, but showing low wettability, was observed for ZnAM50 sample (θav = 83.95°). The application of the Ca-P coating reduced the contact angle (θav = 66.15°)but all the time we have hydrophilic surface. Living organisms react to biomaterials through their surface layer. The analysis of biomaterial surface coating regarding their wettability allows evaluating the possible reaction of organism cells. The ZnAM50 sample surface is hydrophilic. Higher wettability, and thus lower wetting angle, favour cell adhesion to biomaterial surface, which is important in the case of orthopaedic implants as itinfluences the process of tissue – implant integration. On the other hand, lower wettability – and thus a higher value of wetting angle, is vital in the case of such clinical applications as, for example, elements of cardiacvalvesor devices used in dialys is where the lowest protein absorption is desired as it cuts down blood coagulation.

Round 2

Reviewer 2 Report

No other comments

Author Response

The authors would like to say thanks to Reviewer for remarks and suggestions. The authors improve work by eliminate some type errors in text.The changes were marked by using green colour in the revised manuscript. Authors are very grateful for helpful comments in case of the increase of the presented paper readability and quality.

Reviewer 3 Report

Dear authors,

I recommendation accept after minor revision, control some type errors in text.

Author Response

(The authors gave the same response as above.)

Reviewer 4 Report

The authors make all the suggestions and remarks recommended. Considering this I suggest the publication of the manuscript in the Materials Journal as it is.

Author Response

(The authors gave the same response as above.)
